# Active generation of nanoholes in DNA origami scaffolds for programmed catalysis in nanocavities

Jianbang Wang[1], Liang Yue[1], Ziyuan Li[2], Junji Zhang [2], He Tian [2] & Itamar Willner[1]*

DNA origami tiles provide nanostructures for the spatial and temporal control of functional loads on the scaffolds. Here we introduce the active generation of nanoholes in the origami scaffolds using DNAzymes or light as triggers and present the programmed and switchable catalysis in the resulting nanocavities. We engineer "window" domains locked into the origami scaffolds by substrates of the $Zn^{2+}$-ion- or $Pb^{2+}$-ion-dependent DNAzymes. Using $Zn^{2+}$ ions and/or $Pb^{2+}$ ions, the programmed unlocking of the "window" domains is demonstrated. The tailored functionalization of the origami scaffolds allows the programmed operation of catalytic processes in the confined nanocavities. Also, the "window" domain is integrated into the origami scaffold using photoisomerizable azobenzene-modified locks. The cyclic photoisomerization of the locks between the cis and trans states leads to a reversible opening and closure of the nanoholes and to the cyclic light-induced switching of catalytic processes in the nanocavities.

[1] Institute of Chemistry, Center for Nanoscience and Nanotechnology, The Hebrew University of Jerusalem, Jerusalem 91904, Israel. [2] Key Laboratory for Advanced Materials, School of Chemistry and Molecular Engineering, East China University of Science and Technology, Shanghai, China. *email: itamar.willner@mail.huji.ac.il

The programmed assembly of two-dimensional (2D) and three-dimensional (3D) DNA origami nanostructures represents a major advance in DNA nanotechnology[1–3]. Besides ingenious shapes of origami structures generated by the programmed folding of the long-chain M13 phage DNA with dictated "staple" strands, origami structures were functionalized with protruding nucleic acid tethers or edge-modified oligonucleotide strands. The protruding strands were used as anchoring sites for the organization of polymers[4,5], proteins[6], and nanoparticles[7–9] on the origami scaffolds. Unique functions of the nanostructures assembled on the origami scaffolds were demonstrated, such as the operation of enzyme cascades[10,11], the design of plasmonic antennas[12,13], and the assembly of chiroplasmonic structures[14,15]. In addition, dynamic processes of the constituents linked to the origami structures were demonstrated[16]. These included, e.g., the design of DNA walkers[17–19], electric field operation of a robotic arm[20], and the signal-triggered translocation of chiroplasmonic nanostructures[14,21–23]. The edge functionalization of origami tiles was applied to design programmed multi-component origami structures[24] and particularly to develop switchable origami dimers[25–27]. For example, the reversible pH-driven formation of edge-confined, i-motif, or triplex nucleic acid functionalities were applied to stimulate the reversible dimerization and separation of origami tiles. In addition, pH or light was used to induce the reversible isomerization of linear/bent origami nanostructures[25,28]. In addition, the programmed reversible exchange of the compositions of the pairs of origami tiles using the $K^+$-ion-induced formation of G-quadruplexes and their separation, in the presence of crown ether, was demonstrated[29]. Besides 2D DNA origami nanostructures, ingenious 3D origami systems were fabricated. For example, the self-assembly of an origami box[30], the stepwise assembly of gigadalton-scale programmable DNA structures[31], and the light-driven motion of 3D origami bundles to yield reversible chiroptical functions[21,32] have been demonstrated. Different applications of origami nanostructures were suggested, including programmed catalysis[33], controlled drug-release[34,35], logic gate operations[36,37], and sensing[38].

Most of these functional origami structures involved, however, the bottom-up modification of the origami rafts, the edge modification of origami tiles, or the folding of the tiles into tubes. One may, however, consider the functionalization of origami structures with nanocavities (holes or barrels) that might act as containments or channels for guided chemical transformations. To date, such cavities have been fabricated within the passive assembly of the origami tiles[39,40] and these cavities were used for the site-specific docking of antibodies[41], the reconstitution of membrane proteins[42], and the functionalization of solid-state pores for selective transport[40,43]. In addition, DNA structures (not origami) have been introduced into membranes and these acted as channels for the potential-stimulated transport of charges species across the membranes[44,45]. In contrast, the present study introduces the concept of "active" fabrication of nanoholes in origami tiles. We report on the DNAzyme-guided active formation of nanoholes in the origami scaffolds and the molecular "mechanical" unlocking of the nanoholes by lifting the covered "window" domains. By applying two different DNAzymes, the programmed and triggered fabrication of nanoholes in the origami structures is demonstrated. We further utilize the cavities in the different origami scaffolds as confined nano-environments for selective and specific catalysis. In addition, we highlight a design for the reversible light-driven mechanical opening and closure of the nanoholes, and the switchable catalysis in the nanocavities.

## Results

**Active and programmed generation of nanoholes by DNAzymes.** Catalytic nucleic acids (DNAzymes) find broad interest in the area of DNA nanotechnology due to their versatile applications for sensing[46–49], operation of nanodevices[19,50], synthesis of smart materials, e.g., hydrogels[51], and the development of stimuli-responsive drug carriers[52]. For example, the hemin associated with the $K^+$-ion-stabilized G-quadruplex yields a horseradish peroxidase-mimicking DNAzyme[53]. In addition, specific nucleic acid sequences bind metal ions[54–56], e.g., $Mg^{2+}$, $Zn^{2+}$, $Pb^{2+}$, or organic ligands, e.g., histidine[57], to yield supramolecular catalytic structures with nucleic acids as substrates, often ribonucleobase-modified nucleic acid substrates. The catalytic hydrolysis and cleavage of the substrates by these structures were demonstrated. Figure 1a outlines the principles to stimulate the active DNAzyme-driven formation of a nanohole in an origami scaffold. The origami tile N (left in Fig. 1a) includes a domain acting as a "window" to be opened upon the DNAzyme-driven formation of the nanohole in the scaffold. The "window" domain is linked to the origami scaffold by means of eight hinges. In addition, the "window" domain is firmly locked into the origami frame by the crosslinking of the protruding strands $T_1/T_3$ and $T_2/T_4$, linked to the origami frame and the "window" by the hybridization of the strand $L_1$ (complementary sequences "a/a'" and "b/b'"). The interbridging strand $L_1$ includes the domain "z" that provides the substrate for the catalytically inactive $Zn^{2+}$-ion-dependent DNAzyme sequence. In addition, two units ($H_a/H_b$) at opposite sides of the "window", consisting of a single strand linked at its ends to the origami frame (one end) and the "window" raft (the other end), are engineered and act as "handles" assisting the opening of the "window". Furthermore, two protruding anchor strands, $A_1$ and $A_2$, "y-$e_1$'-y" and "y-$e_2$'-y", respectively, are engineered onto the origami tile for stretching the "window" into the open position (vide infra) (Fig. 1a and Supplementary Fig. 1). In the presence of $Zn^{2+}$ ions and the helper hairpins $H_1/H_2$, the "window" domain is opened actively by the following mechanism: the $Zn^{2+}$-ion-dependent DNAzyme[55,58] cleaves the two substrate domains "z" associated with the locks (Fig. 1b). The hairpins $H_1/H_2$ added to the origami system include the domains "x-$e_1$-x-$d_1$-$c_1$"/"$d_2$-$c_2$-x-$e_2$-x" that hybridize with the handles $H_a/H_b$ placed on opposite sides of the "window." It leads to the opening of $H_1$ and $H_2$ to yield the "toehold" tethers "x-$e_1$-x" and "x-$e_2$-x," which form duplexes "$e_1/e_1$'" and "$e_2/e_2$'" with the protruding anchors $A_1$ and $A_2$, respectively. This process provides the hybridization-guided mechanical stretching and opening of the "window" on the origami raft and the generation of the nanohole (right in Fig. 1a). Figure 1c, d show the atomic force microscopy (AFM) images of the origami tiles before (Fig. 1c) and after (Fig. 1d) the addition of $Zn^{2+}$ ions and the helper hairpins $H_1/H_2$. Before the addition of $Zn^{2+}$ ions and $H_1/H_2$, intact origami tiles are observed (Fig. 1c). The addition of $Zn^{2+}$ ions and $H_1/H_2$ leads to the formation of nanoholes in the origami tiles (Fig. 1d). The cross-section analysis of the origami tiles confirms the $Zn^{2+}$ ions/ $H_1/H_2$-guided unlocking of the "windows" and the formation of the nanoholes. Although the intact origami tiles show a height of ca. 2 nm and length of ca. 100 nm, the interaction with $Zn^{2+}$ ions/$H_1/H_2$ yields a domain ca. 50 nm long and 2 nm high, followed by a vacant domain (20–25 nm long) and a high domain (ca. 4 nm high) that corresponds to the open origami "window" laying down on the origami raft (double height of the origami-bright domain on the AFM images). Statistical analyses of four 2 $\mu m \times 2$ $\mu m$ scanned areas reveal that ca. 70% of the origami tiles include the nanoholes, where ca. 30% are either in the locked configuration or not well-defined structures (Fig. 1e, Supplementary Fig. 2 and Supplementary Table 1). It should be noted that both components of the $Zn^{2+}$ ions and $H_1/H_2$

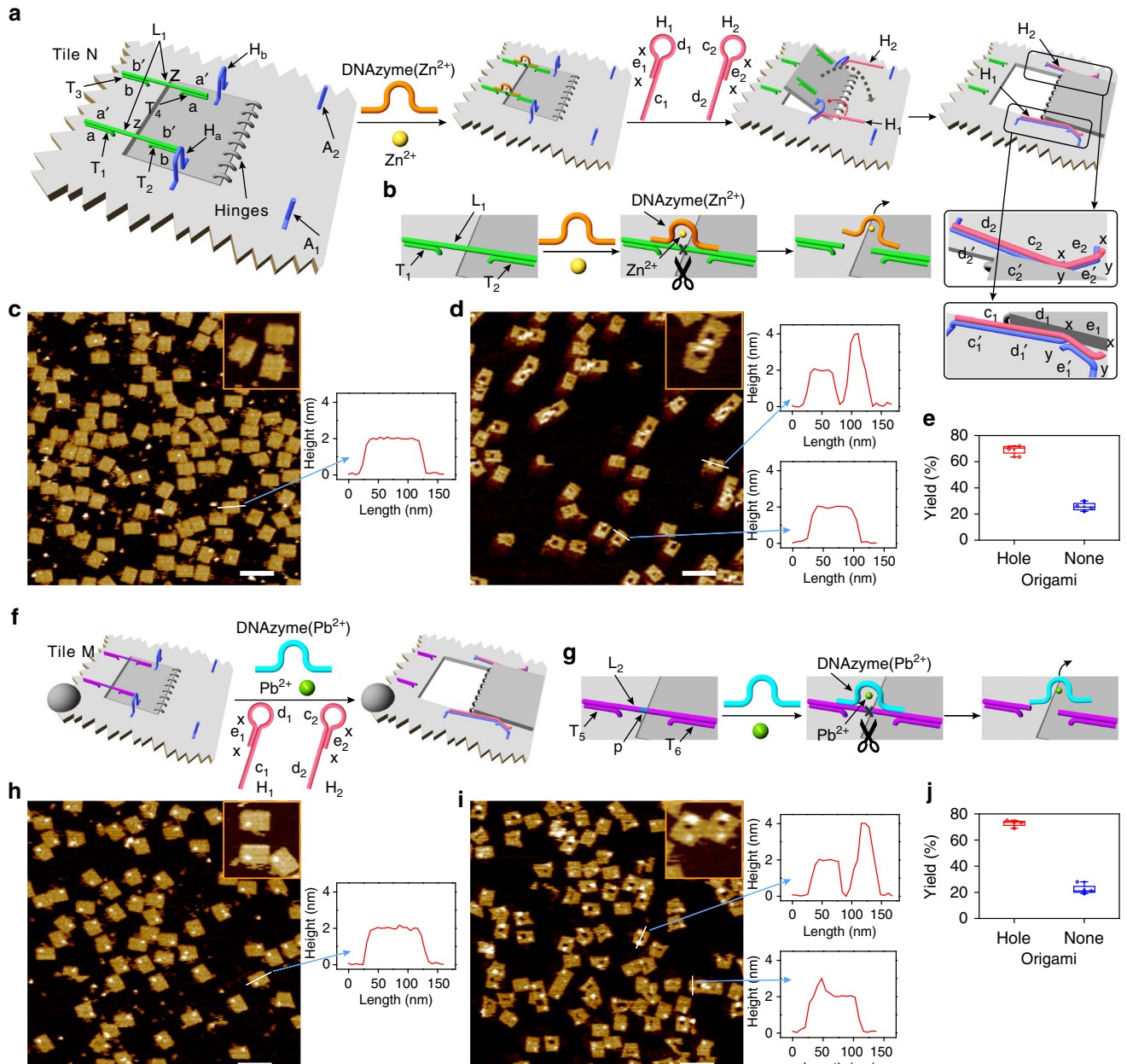

**Fig. 1** DNAzyme-driven formation of nanoholes in origami tiles. **a** Schematic mechanistic formation of the nanohole in the origami tile N using the $Zn^{2+}$-ion-dependent DNAzyme as unlocking catalyst. The origami tile includes a locked "window" domain linked to the origami scaffold by eight hinges, and is modified by two "handles" and two "anchor" tethers. Note that the scheme only shows the key part of the origami tile. **b** Schematic $Zn^{2+}$-dependent DNAzyme-driven unlocking of the "window" by the cleavage of the substrate using the hairpin-assisted opening of the "window" through hybridization to the handles and its pulling to the anchoring sites. **c** AFM image corresponding to the locked configuration of origami tiles. **d** AFM image corresponding to the unlocked origami tiles that include the nanoholes. Insets: enlarged structures of the respective tiles. The AFM images are accompanied by respective cross-section analyses of the tiles confirming the active formation of the nanoholes. Source data are provided as a Source Data file. **e** Statistical analysis of the nanohole-functionalized tiles generated by the $Zn^{2+}$-ion-dependent DNAzyme (for details, see Supplementary Table 1). **f** Schematic mechanistic formation of the nanohole in the origami tile M using the $Pb^{2+}$-ion-dependent DNAzyme as unlocking catalyst. The origami tile is labeled with a four-hairpin label (marked with a hemispheroid) to identify the tile. **g** Schematic $Pb^{2+}$-ion-dependent DNAzyme-driven unlocking of the "window" by the cleavage of the ribonucleobase-modified substrate, associated with the origami tile. **h** AFM image corresponding to the origami tiles prior to the unlocking process. **i** AFM image of the origami tiles after the $Pb^{2+}$-ion-dependent DNAzyme-driven unlocking of the "window". Insets correspond to enlarged structures of the respective tiles. Cross-section analyses confirm the structural features of the labeled nanohole-modified tiles. Source data are provided as a Source Data file. **j** Statistical analysis of the "nanohole"-functionalized tiles generated by the $Pb^{2+}$-ion-dependent DNAzyme (for details, see Supplementary Table 4). Error bars indicate the SD from the analyses of four imaged areas ($2\,\mu m \times 2\,\mu m$). Scale bars: 200 nm

are essential to unlock the "window" and to generate the nanoholes. In the presence of $Zn^{2+}$ ions only, a low yield (ca. 8%) of open nanoholes are observed, and in the presence of $H_1$/ $H_2$, only no nanohole-containing origami structures are detected

(Supplementary Figs. 3 and 4, and Supplementary Tables 2 and 3; in addition, Supplementary Fig. 5 provides a statistical analysis of the results according to Student's *T*-test). Further support for the DNAzyme-triggered unlocking of the "window" domain, and the

subsequent "helpers" induced opening of the "window" and its fixation on the anchoring sites, was obtained by complementary fluorescence resonance energy transfer (FRET) experiments. In these experiments, the handle $H_a$ was internally modified with Cy3, and the anchoring foothold was modified with Cy5. The FRET signal (Cy3 to Cy5) generated upon the "helper"-induced opening of the window, and the use of an appropriate calibration curve allowed us to quantify the yield of "open window" structures (for a detailed discussion of the FRET experiments, extraction of the calibration curve, the FRET results, and the quantification of the "open window" origami structures, see Supplementary Fig. 6 and accompanying discussion). The results reveal that the yield of "open window" in origami structure is ca. 70%, in agreement with the statistical analysis of AFM images.

Similarly, the active unlocking of the "window" domain associated with the origami scaffold M using the $Pb^{2+}$-ion-dependent DNAzyme[56] as the unlocking catalyst was demonstrated (Fig. 1f). In this system, the lock strand $L_2$ is used to crosslink the "window" domain to the origami scaffold (Fig. 1f, g). The strand $L_2$ includes the ribonucleotide-containing domain "p" that acts as the substrate of the $Pb^{2+}$-ion-dependent DNAzyme. The tile M is modified with the eight hinges, the two handle units, and the protruding tethers for anchoring of the open "window" through hybridization of the helpers/handles/anchors (Supplementary Fig. 7). The $Pb^{2+}$-ion-dependent DNAzyme sequence is hybridized, in a catalytically inactive configuration, with the substrate sequence "p". In the presence of $Pb^{2+}$ ions and $H_1/H_2$, the substrate domains are cleaved (Fig. 1g), leading to the active unlocking of the "window" and to the opening of the nanohole by the "handle"-stimulated opening of $H_1/H_2$ and the "mechanical" stretching of the "window" on the origami raft by hybridization of the single-stranded toehold associated with the open hairpins with anchoring sites $A_1/A_2$. It is noteworthy that the origami tile unlocked by the $Pb^{2+}$-ion-dependent DNAzyme is labeled with a four-hairpin label (to distinguish between the origami tiles being unlocked by $Zn^{2+}$ ions and/or $Pb^{2+}$ ions, vide infra). Figure 1h, i show the AFM images corresponding to the intact locked origami tiles and the unlocked origami tiles, respectively. The label (bright spot) is observed on all tiles. Subjecting the tiles to $Pb^{2+}$ ions and $H_1/H_2$ unlocks the "window" domains leading to the opening of the nanoholes (inset shows the enlarged image of the resulting tiles). In addition to the hole and the label, the folding of the "window" across the "hinges" and the lay-down of the "window" on the origami raft opposite to the locking domain are clearly visible (brighter and higher domain). The cross-section analysis of the respective tiles shows the intact locked tiles with the characteristics ca. 100 nm length and ca. 2 nm height. A spike ca. 1 nm high corresponding to the label is observed on the background height of the origami frame. The unlocked tiles show the background height of the origami scaffold, 2 nm, a void domain that confirms the formation of the nanoholes, and a domain of double height, ca. 4 nm, consistent with the lay-down of the stretched "window" on the origami. Figure 1j presents the statistical analysis of the contents of the unlocked nanohole-origami tiles (four 2 μm × 2 μm scanned areas; Supplementary Fig. 8 and Supplementary Table 4), where ca. 72% of the origami tiles include the nanoholes. As before, control experiments imply that the activation of the $Pb^{2+}$-ion-dependent DNAzyme and the addition of $H_1/H_2$ are essential to generate the high yield of nanoholes (see Supplementary Figs. 9 and 10, and Supplementary Tables 5 and 6, and accompanying discussion). These results indicate the high-yield formation of nanohole-containing origami structures upon the $Pb^{2+}$-ion-dependent DNAzyme cleavage of the locks and the concomitant "mechanical" stretching of the "window" on the origami rafts by

means of the handles/helper hairpins/anchoring footholds ($H_a/H_1/A_1$ and $H_b/H_2/A_2$).

The programmed unlocking of the locked origami tiles, in the presence of $Zn^{2+}$ ions and/or $Pb^{2+}$ ions, was then evaluated (Fig. 2). In the presence of $Zn^{2+}$ ions and the helper hairpins, only the origami tiles lacking the labels, N, are unlocked to yield the nanoholes. In the presence of $Pb^{2+}$ ions and the helper hairpins, only the labeled origami tiles, M, are unlocked to yield the nanoholes, and in the presence of $Zn^{2+}$ ions, $Pb^{2+}$ ions and the helper hairpins, the guided formation of the nanoholes in the two origami tiles proceeds (Fig. 2a). Figure 2b-e show the AFM images of the respective systems and the statistical analysis of the contents of the tiles in the different systems. Figure 2b shows the AFM image of the intact origami tiles N and M, and Fig. 2c depicts the AFM image of the origami mixture after treatment with $Zn^{2+}$ ions and $H_1/H_2$. The unlabeled tiles N reveal a high content of nanohole-modified rafts, whereas no nanoholes are observed in the labeled origami tiles M. The statistical analysis of the origami tiles indicates ca. 70% of hole-containing rafts and ca. 30% of closed or non-defined structures of the unlabeled tiles N. The content of the closed tiles M is ca. 100% (Supplementary Fig. 11 and Supplementary Table 7). Figure 2d shows the AFM image of the origami mixture treated with $Pb^{2+}$ ions and $H_1/H_2$. The unlabeled tiles N stay closed (100%), whereas the labeled tiles M include nanoholes (ca. 72%) and a content of ca. 28% origami rafts that stay locked or show a non-defined structure (respective statistical evaluation of the structures; Supplementary Fig. 12 and Supplementary Table 8). Upon treatment of the origami mixture with $Zn^{2+}$ ions, $Pb^{2+}$ ions, and $H_1/H_2$, both tiles, N and M, reveal nanohole-containing structures with yields corresponding to ca. 72% and ca. 75%, respectively (Fig. 2e, Supplementary Fig. 13, and Supplementary Table 9). It should be noted that the hairpin marker in tile M is distinguishable from the "open-window" domain associated with tile N by the height characterizing the hairpin marker (3 nm) and the height of a double-layer origami structure of the "open window" domain (4 nm).

**Programmed catalysis in DNAzyme-drilled nanocavities.** In the next step, the $Zn^{2+}$ ions/$Pb^{2+}$ ions programmed unlocking of the origami tiles were used to stimulate controlled catalysis in the confined nanocavities (Fig. 3). Towards this goal, the $Zn^{2+}$-ion-responsive tile N is further engineered to generate the tile $N_f$ (Top-middle and left enlargement of Fig. 3a and Supplementary Fig. 14). One side of the origami raft is modified with the strand $K_{his}$ that hybridizes with the protruding strands $T_{N1}/T_{N3}$. The bottom side of the origami raft is functionalized with the hairpin $E_{his}$ that is attached to the raft through hybridization with the protruding strands $T_{N2}/T_{N4}$. It is noteworthy that the origami raft domain is engineered with $2 \times K_{his}/E_{his}$ units. The hairpin $E_{his}$ includes the histidine-dependent DNAzyme sequence in a caged, inactive, configuration. The locked structure of the origami tile prohibits inter-communication between $K_{his}$ and $E_{his}$. The $Zn^{2+}$-ion-induced unlocking of the "window" allows the interaction between $K_{his}$ and $E_{his}$ in the resulting cavity (middle-left in Fig. 3a). The units $K_{his}$ and $E_{his}$ are predesigned to allow the $K_{his}$-induced opening of the hairpin $E_{his}$ to yield the uncaged histidine-dependent DNAzyme[57] in the cavity. The binding of the histidine-dependent DNAzyme to the FAM/BHQ1-modified substrate S leads, in the presence of histidine, to the cleavage of S. The resulting fluorescence of the FAM-modified fragment provides the transduction signal for the catalytic process proceeding in the confined cavity of the $Zn^{2+}$-ion-responsive tiles $N_f$ (Supplementary Fig. 15). Similarly, top-middle and accompanying right enlargement of Fig. 3a depict the modification of the origami tile M to form tile $M_f$ with an engineered cavity allowing

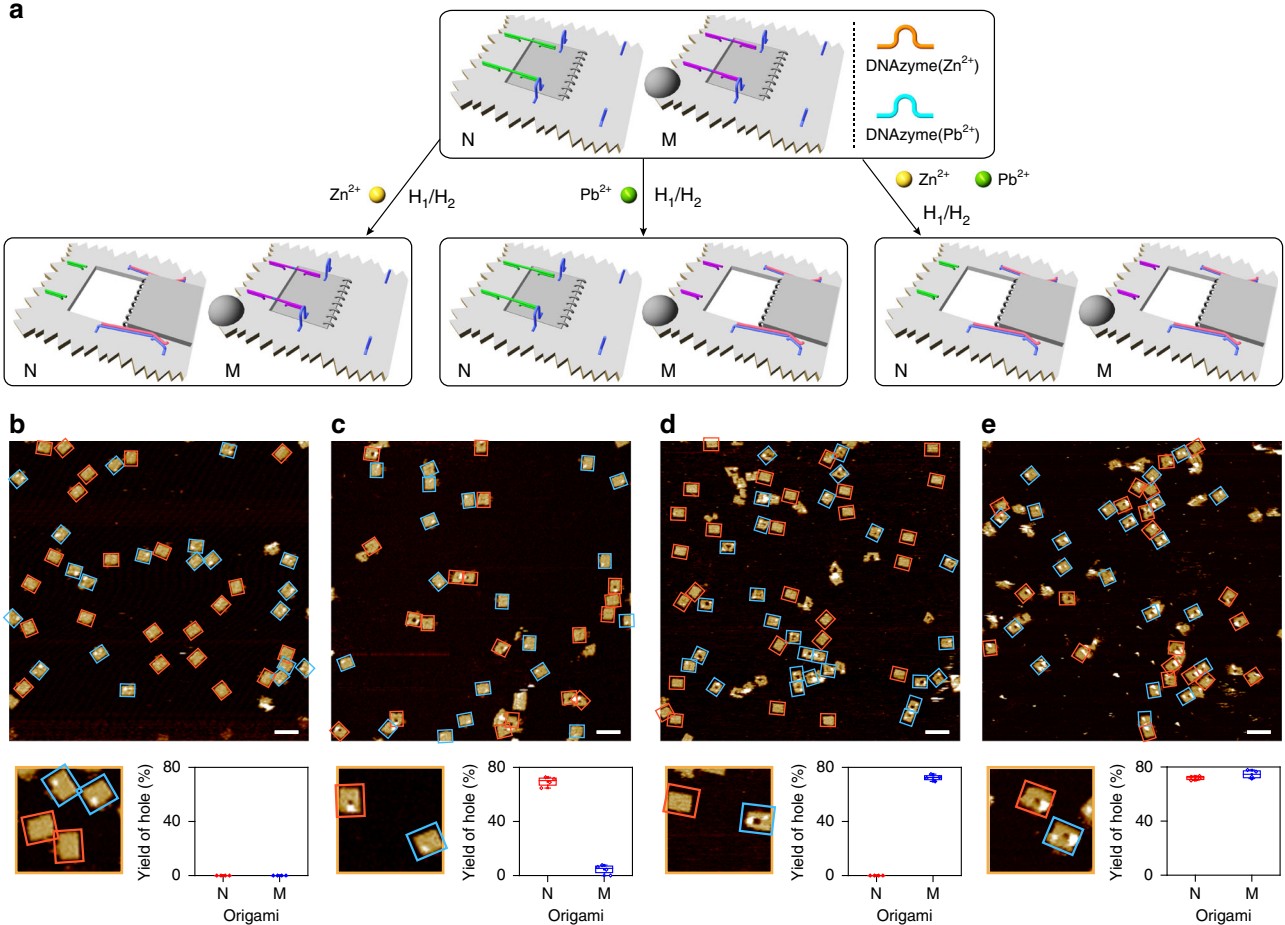

**Fig. 2** Programmed and selective formation of nanoholes in the mixture of origami tiles N and M, dictated by the two DNAzymes. **a** Top, schematic mixture of the origami tiles N and M. Bottom-left, selective unlocking and formation of nanoholes in tiles N by activating the $Zn^{2+}$-ion-dependent DNAzyme only. Bottom-middle, selective unlocking and formation of nanoholes in origami tiles M by activating the $Pb^{2+}$-ion-dependent DNAzyme only. Bottom-right, the concomitant formation of nanoholes in the mixture of origami tiles N and M upon activation of the two $Zn^{2+}$-ion- and $Pb^{2+}$-ion-dependent DNAzymes. **b** AFM image corresponding to the mixture of locked origami tiles N and M. **c** AFM image of the origami tile mixture generated upon the activation of the $Zn^{2+}$-ion-dependent DNAzyme (Cf. bottom-left in **a**). **d** AFM image corresponding to the origami tile mixture generated upon the activation of the $Pb^{2+}$-ion-dependent DNAzyme (Cf. bottom-middle in **a**). **e** AFM image of the origami tile mixture generated upon the activations of $Zn^{2+}$-ion- and $Pb^{2+}$-ion-dependent DNAzymes (Cf. bottom-right in **a**). Each of the AFM images is accompanied by a statistical analysis of the origami constituents N and M in the respective systems (for details, see Supplementary Tables 7–9). Error bars indicate the SD from the analyses of four imaged areas (2 μm × 2 μm). Scale bars: 200 nm

the programmed activation of the hemin/G-quadruplex horseradish peroxidase-mimicking DNAzyme (Supplementary Fig. 16). Toward this end, the upper origami raft is functionalized with the guanosine-rich sequences $G_1$ that are linked to the raft through hybridization with the protruding tethers $T_{M1}/T_{M3}$. The bottom surface of the origami is functionalized by the guanosine-rich sequences $G_2$ that are linked to the raft through hybridization with the protruding tethers $T_{M2}/T_{M4}$. It is worth noting that the origami raft is modified with $2 \times G_1/G_2$ pairs. In the locked configuration, the strands $G_1$ and $G_2$ are separated by the origami raft and cannot interact. The $Pb^{2+}$-ion-induced unlocking of the origami tile allows the inter-communication of $G_1$ and $G_2$ in the resulting cavity. In the presence of $K^+$ ions and hemin, the two strands assemble into the $K^+$-ion-stabilized hemin/G-quadruplex horseradish peroxidase-mimicking DNAzyme that catalyzes the $H_2O_2$ oxidation of Amplex Red to the fluorescent Resorufin product. The resulting fluorescence of Resorufin provides a transduction signal for the catalytic process proceeding in the confined cavity of the $Pb^{2+}$-ion-responsive tiles $M_f$ (middle-center of Fig. 3a and Supplementary Fig. 17). Figure 3b shows the

programmed activation of the DNAzyme-catalyzed processes operating in the respective nanocavities of the origami tiles (Supplementary Fig. 18). In these experiments, the mixture of the tiles $N_f$ and $M_f$ is subjected to the respective ion triggers. The mixture in the absence of $Zn^{2+}$ ions or $Pb^{2+}$ ions does not show any fluorescence signals (entry I) consistent with the lack of formation of any active DNAzymes. In the presence of $Zn^{2+}$ ions, only the fluorescence of FAM is observed, and no fluorescence of Resorufin is detected (entry II). This is consistent with the selective $Zn^{2+}$-ion-stimulated activation of the histidine-dependent DNAzyme units associated with the tile $N_f$. Similarly, subjecting the mixture to $Pb^{2+}$ ions results in the selective unlocking of tile $M_f$, accompanied by the generation of the fluorescence of Resorufin (entry III). Under these conditions, no fluorescence of FAM is observed, indicating that the tiles $N_f$ stay in the locked, inactive, configuration. Treatment of the mixture with $Zn^{2+}$ ions and $Pb^{2+}$ ions leads to the unlocking of the two types of origami tiles, resulting in the fluorescence of FAM and Resorufin (entry IV) consistent with the activation of the two types of DNAzymes. An important issue to address involves,

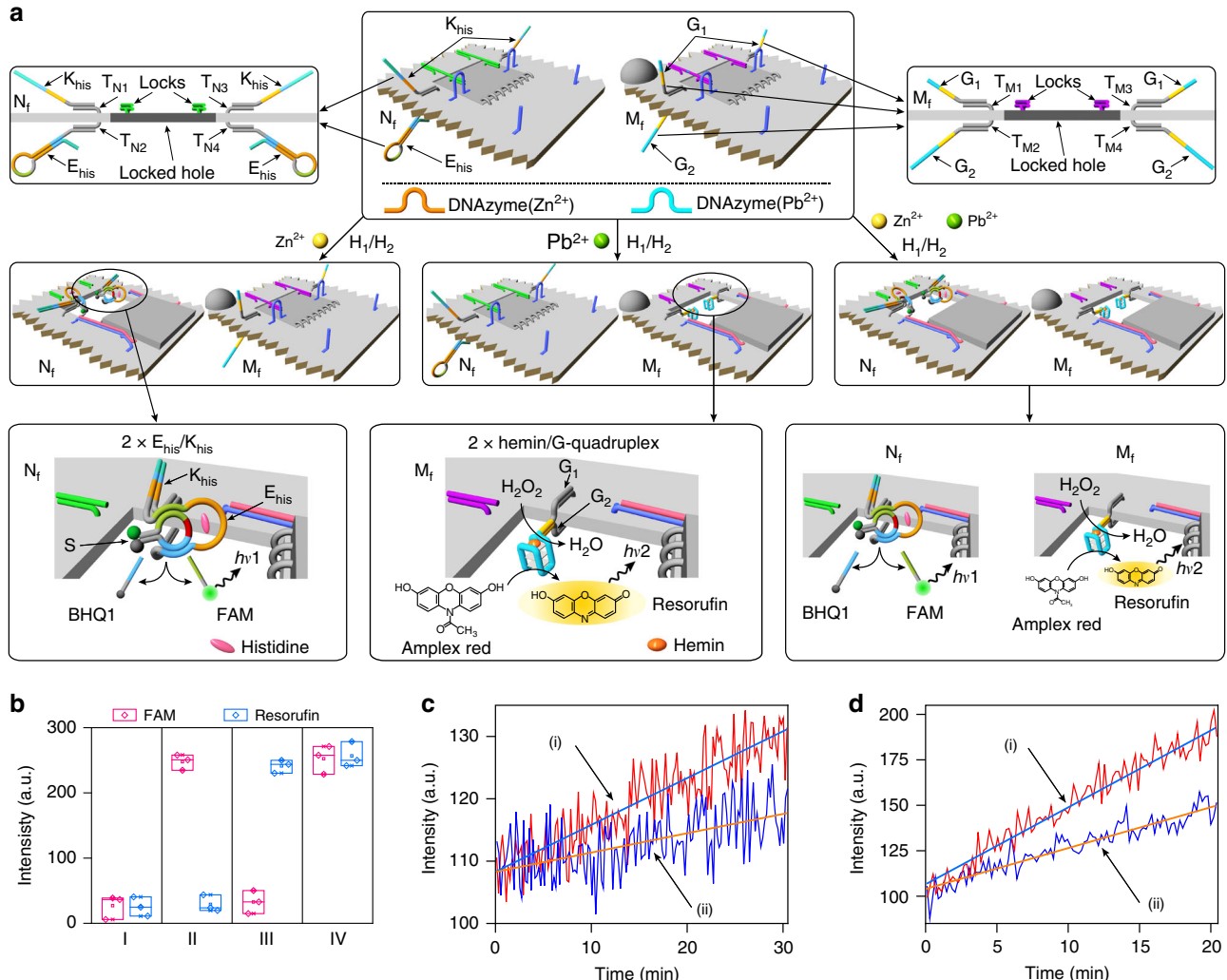

**Fig. 3** Programmed catalytic transformation in confined cavities in DNA origami scaffolds. **a** The mixture of two origami tiles $N_f$ and $M_f$, which include the locking/unlocking components for the opening of respective "window" domains by the $Zn^{2+}$-ion- and/or $Pb^{2+}$-ion-dependent DNAzymes (Top). Middle-left, treatment of the origami mixture with $Zn^{2+}$ ions unlocks the "window" associated with tile $N_f$ and its opened "window" allows the $K_{his}$ strand-induced uncaging of the $E_{his}$ and the activation of the histidine-dependent DNAzyme in the resulting cavity. Middle-center, subjecting the origami mixture to $Pb^{2+}$ ions leads to the unlocking of the "window" of tile $M_f$. The opened "window" in tile $M_f$ allows the interaction of the subunits $G_1$ and $G_2$, and the assembly of the hemin/G-quadruplex DNAzyme in the resulting cavity. Middle-right, treatment of the origami mixture with $Zn^{2+}$ ions and $Pb^{2+}$ ions leads to the unlocking of the two "windows" associated with $N_f$ and $M_f$. The opened "windows" in both tiles allow the activation of the respective catalytic transformations. **b** Fluorescence outputs (FAM and/or resorufin) generated by: Entry I, the locked mixture of $N_f$ and $M_f$. Entry II, the treatment of the origami mixture with $Zn^{2+}$ ions and the activation of the histidine-dependent DNAzyme in the confined cavity of $N_f$. Entry III, the treatment of the origami mixture with $Pb^{2+}$ ions and the activation of the hemin/G-quadruplex DNAzyme in the confined cavity of $M_f$. Entry IV, treatment of the origami mixture with $Zn^{2+}$ ions and $Pb^{2+}$ ions. Error bars indicate the SD from three independent experiments (for details, see Supplementary Fig. 18). **c** The catalytic activities of the histidine-dependent DNAzymes in: (i) the confined cavities in the origami tiles $N_f$ and (ii) the homogenous identical buffer solution. The concentration of the histidine-dependent DNAzymes in the cavity or bulk solution is identical, 24.6 nM. Source data are provided as a Source Data file. **d** The catalytic activities of the hemin/G-quadruple DNAzymes in: (i) the confined cavities in the origami tiles $M_f$ and (ii) the homogenous identical buffer solution. The concentration of the hemin/G-quadruple DNAzymes in the cavity or bulk solution is identical, 17.9 nM. Source data are provided as a Source Data file

however, an assessment on the activities of the hemin/G-quadruplex and histidine-dependent DNAzyme in the confined reaction cavities in the origami tiles vs. the activities of these DNAzymes in a homogenous aqueous phase (under the same concentrations). In fact, previous studies demonstrated that biocatalytic cascades in confined environments reveal enhanced activities due to the spatial concentrations of the catalysts in the confined volumes[59–61]. Thus, the confinement of the different DNAzyme subunits could affect the activities of the DNAzymes. Knowing the concentrations of the active DNAzyme associated with the cavities, we compared their activities in the confined

environments with the activities of the DNAzyme in a homogenous phase at the same concentrations (Figs. 3c, d). We find that the activities of the histidine-dependent DNAzyme and of the hemin/G-quadruplex horseradish peroxidase-mimicking DNAzyme in the confined cavities are ca. twofold higher as compared with their activities in the homogenous solution.

**Light-induced switchable unlocking and locking of nanoholes**. The DNAzyme-catalyzed formation of the nanoholes in the origami tiles and the resulting catalytic reactions in the confined

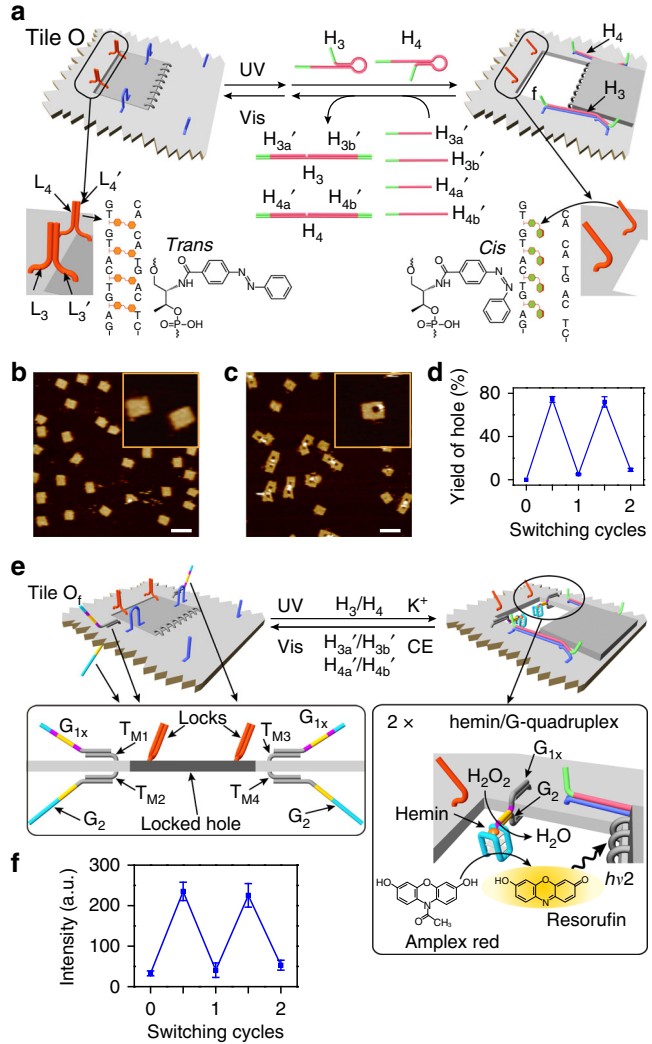

**Fig. 4** Switchable photoinduced unlocking and locking of the cavity in the origami tile. **a** Mechanism for the photoinduced unlocking and locking of the "window" domain in the origami tile O using trans-azobenzene stabilized locks. Photoisomerization of the trans-azobenzene units to cis-azobenzene separates the locks and the $H_3/H_4$ assisted mechanical opening of the nanoholes. Subjecting the "opened window" to the counter strands and the photoisomerization of the cis-azobenzene units to trans-azobenzene, recovers the locked configuration of tile O. **b** AFM image corresponding to the locked origami tiles. **c** AFM image of the "opened window" origami tiles generated upon photochemical unlocking of the tiles and concomitant treatment with $H_3/H_4$. The reverse locking of the tiles is achieved by irradiation of the "opened window" tiles and concomitant treatment of the tile with anti-helper strands. **d** Statistical analysis of the "open window" tile upon the cyclic irradiation of the origami tile in the presence of the respective helper hairpins and the anti-helper strands, respectively (for details, see Supplementary Tables 10–14). Error bars indicate the SD from the analyses of four imaged areas (2 μm × 2 μm). **e** Mechanism for the photoinduced reversible activation of the hemin/G-quadruplex DNAzyme in the photogenerated cavity associated with the tile $O_f$. The photoinduced unlocking of the "window" and in the presence of hairpins $H_3/H_4$, and $K^+$ ions and hemin lead to the formatioin of the hemin/G-quadruplex DNAzyme in the resulting cavity, resulting in catalyzed $H_2O_2$ oxidatioin of Amplex Red to the fluorescent Resorufin product. Irradiation of the "open window"/catalytic system in the presence of the anti-helper strands and 18-crown-6-ether results in the separation of the G-quadruplex catalytic units and the regeneration of the locked configuration of the tile. **f** Cyclic and reversible fluorescence changes generated by the origami tiles upon photoinduced unlocking or locking the tiles in the presence of the respective helpers/anti-helper strands and added $K^+$ ions/crown ether. Error bars indicate the SD from three independent experiments (for details, see Supplementary Fig. 30). Scale bars: 200 nm

anchoring tethers, $A_1/A_2$, leading to the formation of $H_3/H_{3a}'/H_{3b}'$ and $H_4/H_{4a}'/H_{4b}'$ as wastes and to the reclosure of the "flexible window" into the locked structure. By the reversible photochemical trans ⇔ cis isomerization of the azobenzene units in the presence of the helper and counter-helper strands, the nanoholes are cycled between open and closed states, respectively. Figure 4b, c show the AFM images of the intact closed tiles and the photogenerated nanohole-containing tiles (λ = 365 nm) in the presence of the helper hairpins, respectively. Figure 4d shows the cyclic opening and closure of the nanoholes upon the photo-isomerization of the azobenzene units in the presence of the helper and counter-helper strands. The yield of nanohole-containing tiles is ca. 74% (Supplementary Figs 20–24 and Supplementary Tables 10–14). As before, the photoisomerization of the azobenzene units (λ = 365 nm) and the added hairpins $H_3/H_4$ are essential to generate the high yield of the stretched open "window" (see Supplementary Figs. 25–27 and Supplementary Tables 15 and 16, and accompanying discussion).

**Light-induced switchable catalysis in nanocavities.** The light-induced opening and closure of the nanoholes were then applied to reversibly switch the hemin/G-quadruplex horseradish peroxidase-mimicking DNAzyme in the photogenerated nano-cavities (Fig. 4e and Supplementary Fig. 28). Toward this goal, the upper and lower surfaces of the photo-responsive origami tiles are functionalized with guanosine-rich strands $G_{1x}$ and $G_2$, which hybridize with the protruding tethers $T_{M1}/T_{M3}$ and $T_{M2}/T_{M4}$, respectively. In the locked configuration, the strands $G_{1x}$ and $G_2$ are separated by the origami tile (left of Fig. 4e). The photo-chemical unlocking of the "window" and its opening in the presence of $K^+$ ions and the helper hairpins result in the

cavities are, however, single-cycle processes and the switchable locking and unlocking of the nanoholes are practically prohibited. We, thus, searched for a locking/unlocking trigger that excludes the need to add nucleic acid strands. This is feasible by the application of light signals to reversibly open and close the nanoholes. This was achieved by applying photoisomerizable azobenzene-modified oligonucleotides as light-triggered units for the reversible unlocking and locking of the nanoholes as outlined in Fig. 4a. The "window" is engineered into the origami tile O by applying eight hinges and two handles, $H_a/H_b$, on opposite sides of the "window," which link the "window" to the origami raft. The "window" is locked to the raft through the formation of two duplex locks between protruding strands $L_3/L_3'$ and $L_4/L_4'$. The locked duplexes are stabilized by trans-azobenzene units that intercalate in base pairs. In addition, the origami tile is modified with two protruding foothold tethers, $A_1/A_2$, which function as anchoring sites for stretching and fixing of the open "window" (Supplementary Fig. 19). Photoisomerization of the trans-azobenzene and in the presence of the helper hairpins, $H_3/H_4$, leads to the unlocking of the "window" and to its "mechanical" opening and stretching over the origami tile by the duplexes $H_a/H_3/A_1$ and $H_b/H_4/A_2$. Subjecting the hole-containing origami tiles to the anti-strands, $H_{3a}'/H_{3b}'$ and $H_{4a}'/H_{4b}'$, and the concomitant photoisomerization of the cis-azobenzene units to trans-azobenzene (λ > 420 nm) result in the displacement of the stretching strands from the handle units, $H_a/H_b$, and the

inter-communication of $G_{1x}$ and $G_2$ positioned on opposite sides of the origami raft and their assembly into the $K^+$-ion-stabilized G-quadruplex in the nanocavity. It is noteworthy that the origami raft is modified with $2 \times G_{1x}/G_2$ pairs that yield two G-quadruplex units in the unlocked cavity. The binding of hemin to the G-quadruplex yields the hemin/G-quadruplex DNAzyme that catalyzes the $H_2O_2$ oxidation of Amplex Red to the fluorescent Resorufin product (right of Fig. 4e). The subsequent treatment of the catalytic origami nanostructure with 18-crown-6-ether (CE), the counter strands $H_{3a}'/H_{3b}'$ and $H_{4a}'/H_{4b}'$, and the illumination of the system with visible light results in the separation of the G-quadruplex DNAzyme (via elimination of the $K^+$ ions, the removal of the stretching strands through the separation of the duplexes $H_a/H_3/A_1$ and $H_b/H_4/A_2$, and the locking of the nanohole through the regeneration of the trans-azobenzene-stabilized duplex locks. That is, the reversible treatment of the origami tiles with UV/strands/$K^+$ ions and visible light/counter strands/crown ether (CE) leads to the cyclic "ON"/"OFF" switching of the hemin/G-quadruplex DNAzyme (Supplementary Fig. 29). Figure 4f demonstrates that the photo-responsive origami tiles indeed activate the switchable "ON"/"OFF" catalytic activities of the DNAzyme by the sequential triggered "mechanical" opening and closure of the cavity in which the catalysts are formed and separated (Supplementary Fig. 30), respectively.

## Discussion

The present study has introduced a method for the active generation of nanoholes in origami tiles and the programmed operation of catalytic transformations in the resulting nanocavities. One approach has included the unlocking of the nanoholes in the origami tiles using metal-ion-dependent DNAzymes as unlocking biocatalysts. The programmed unlocking of a mixture of origami tiles by $Zn^{2+}$-ion- and/or $Pb^{2+}$-ion-dependent DNAzymes, and the subsequent programmed operation of catalytic transformations in the confined nanocavities have been demonstrated. A second method to reversibly unlock and lock the nanoholes by means of auxiliary light signals, and to switch "ON"/"OFF" catalytic functions in the nanocavities using photoisomerizable azobenzene-modified strands as locks were accomplished. Beyond the nanotechnological impact of this study to design nanoholes in origami tiles as confined cavities for programmed catalytic transformations, the results highlight further applications of these nanostructures. For example, by appropriate design of the active origami rafts, the programmed unlocking of different-sized nanohole patterns may be envisaged. Such nanohole arrays could be used for multiplexed sensing. In addition, the incorporation of different enzymes or plasmonic nanoparticles into the engineered nanocavities could provide versatile means to operate biocatalytic cascades or to assemble plasmonic devices.

## Methods

**Materials**. Oligonucleotides were purchased from Integrated DNA Technologies and FAM/BHQ1-labeled oligonucleotide was purified by high-performance liquid chromatography. Single-stranded M13mp18 DNA was obtained from New England Biolabs. Freeze 'N Squeeze DNA Gel Extraction spin columns were purchased from Bio-Rad. Amicon centrifugal filters (100 k, NMWL) were purchased from Merck Millipore Ltd. Other chemical reagents were purchased from Sigma-Aldrich.

**Preparation of DNA origami tiles and the hairpins**. To prepare DNA origami tiles, single-stranded M13mp18 phage DNA (10 nM) and short staple strands (100 nM, unmodified staple strands and functional staple strands) (Supplementary Tables 17–24) were dissolved in the TAE buffer (Tris, 20 mM; acetic acid, 20 mM; EDTA, 1 mM; pH 8.0) with 12.5 mM $Mg^{2+}$. The mixture was heated to 95 °C in a thermal cycler and then allowed to cool down to 20 °C at a rate of $-0.6$ °C min$^{-1}$. The DNA origami tiles were purified using agarose electrophoresis (1%, 100 V,

1.5 h, at 0 °C) to remove the excess staple strands and were then extracted from the gel bands using Freeze 'N Squeeze spin columns.

All the hairpin strands (10 μM) were annealed from 90 °C to 10 °C at a rate of $-3$ °C min$^{-1}$ in the TAE buffer (Tris, 20 mM; acetic acid, 20 mM; EDTA, 1 mM; pH 8.0) with 6 mM $Mg^{2+}$ and 5 mM $Na^+$.

**Formation of nanoholes in the origami tiles N and M**. The purified origami tile N (2 nM) including the $Zn^{2+}$-ion-dependent DNAzyme sequence (20 nM) was added with the $Zn^{2+}$ ions (5 mM) (10 mM Tris, 20 mM $Mg^{2+}$, pH 7.0). The sample was kept at 30 °C for 10 min. Then the sample was centrifuged (100 k NMWL, $3000 \times g$, 10 min, three times) to remove the DNAzyme sequence and the $Zn^{2+}$ ions and the buffer was changed to TAE buffer with 6 mM $Mg^{2+}$ and 5 mM $Na^+$. The sample was added with the helper hairpins $H_1/H_2$ (10 nM for each one) and was kept at 25 °C for 10 h.

For the control experiments on the tile N (2 nM), only $Zn^{2+}$ ions (5 mM) or the helper hairpins $H_1/H_2$ (10 nM for each one) were used in the nanohole-forming processes.

For the FRET measurement, the origami tile N with the internally Cy3-modified handle $H_{a-C3}$ and Cy5-modified anchoring foothold $A_{1-C5}$ was prepared. After the unlocking of the "window" by the $Zn^{2+}$-ion-dependent DNAzyme ($Zn^{2+}$, 5 mM) and the centrifugation (100 k NMWL, $3000 \times g$, 10 min, three times), the fluorescence spectrum of the origami tile N (set at 20 nM) was measured ($\lambda_{ex} = 532$ nm). Then the helper strands $H_1$ and $H_2$ were added into the sample to fix the "window" to the anchoring sites. The fluorescence spectrum of the origami tile N in the open state (set at 20 nM) was measured ($\lambda_{ex} = 532$ nm). For the calibration curve, the mixtures of the strands $H_{a-C3}$ (20 nM) and $A_{1-C5}$ (20 nM) were subjected to $H_{1-C}$ with variable concentrations (0, 5, 10, 15, and 20 nM) and their fluorescence spectra were measured ($\lambda_{ex} = 532$ nm).

To form a nanohole in origami tile M (2 nM), the sample of the purified tile M (2 nM) including the $Pb^{2+}$-ion-dependent DNAzyme sequence (20 nM) was added with $Pb^{2+}$ ions (100 μM) (10 mM Tris, 20 mM $Mg^{2+}$, pH 7.0). The sample was kept at 30 °C for 10 min and then was centrifuged (100 k NMWL, $3000 \times g$, 10 min, three times) to remove the DNAzyme sequence and the $Pb^{2+}$ ions, and the buffer was changed to TAE buffer with 6 mM $Mg^{2+}$ and 5 mM $Na^+$. The sample was added with the helper hairpins $H_1/H_2$ (10 nM for each one) and was kept at 25 °C for 10 h.

For the control experiments on the tile M (2 nM), only $Pb^{2+}$ ions (100 μM) or the helper hairpins $H_1/H_2$ (10 nM for each one) were used in the nanohole-forming processes.

For the selective formation of nanoholes in the origami mixture, the equal amount of purified origami tiles N and M (2 nM for each tile) were mixed with the $Zn^{2+}$-ion-dependent DNAzyme sequence (20 nM) and the $Pb^{2+}$-ion-dependent DNAzyme sequence (20 nM) (10 mM Tris, 20 mM $Mg^{2+}$, pH 7.0). It was divided into four samples and were added with different ion triggers. One sample was the prior mixture without added triggers as a reference. The second sample was added with the trigger of $Zn^{2+}$ ions (5 mM). The third sample was added with the trigger of $Pb^{2+}$ ions (100 μM). The fourth sample was added with triggers of $Zn^{2+}$ ions (5 mM) and $Pb^{2+}$ ions (100 μM). The samples were kept at 30 °C for 10 min, and then the DNAzyme sequences and the metal ions were removed by centrifugation (100 k NMWL, $3000 \times g$, 10 min, three times) and the buffer solutions were changed to TAE buffer with 6 mM $Mg^{2+}$ and 5 mM $Na^+$. Then the helper hairpins $H_1/H_2$ (20 nM for each one) were added and the sample was kept at 25 °C for 10 h.

**Measurement of the programmed catalytic activities**. For the measurements of the catalytic reactions in the origami mixture, four samples of the mixture of the origami tiles $N_f$ and $M_f$ (20 nM, 120 μL for each one) with the $Zn^{2+}$-ion-dependent DNAzyme sequence (200 nM) and the $Pb^{2+}$-ion-dependent DNAzyme sequence (200 nM) (10 mM Tris, 20 mM $Mg^{2+}$, pH = 7) were prepared. Then the four samples were added with different components of $Zn^{2+}$ ions (5 mM) and/or $Pb^{2+}$ ions (100 μM) as triggers to unlock the "windows." One sample was added with no trigger acting as a reference, the second sample was added with $Zn^{2+}$ ions, the third sample was added with $Pb^{2+}$ ions, and the fourth sample was added with both ions. All the samples were kept at 30 °C for 10 min. After removing the DNAzyme sequences and metal ions (100 k NMWL, $3000 \times g$, 10 min, three times), the helper hairpins $H_1/H_2$ (200 nM for each one) were added into the four samples, respectively, and the samples were kept at 25 °C for 10 h in TAE buffer with 6 mM $Mg^{2+}$ and 5 mM $Na^+$. Then each sample was divided into two parts with same volumes. One part was added with histidine (5 mM) and substrate S (1 μM). The concentration of each tile was set at 15 nM (50 μL) in TAE buffer (with 6 mM $Mg^{2+}$, 5 mM $Na^+$, and 200 mM $K^+$). The catalytic reaction was taken at 30 °C for 6 h and the fluorescence spectrum of the product (FAM-labeled fragment) was measured using a Cary Eclipse Fluorescence Spectrophotometer (Varian, Inc.) ($\lambda_{ex} = 495$ nm). The other one was treated with hemin (30 nM), Amplex Red (100 μM) and $H_2O_2$ (5 mM), and the concentration of each tile was 15 nM (50 μL) in TAE buffer (with 6 mM $Mg^{2+}$, 5 mM $Na^+$, and 200 mM $K^+$). After incubation time of 10 min at 28 °C, the fluorescence spectrum of the product (Resorufin) was measured using the fluorescence spectrophotometer ($\lambda_{ex} = 571$ nm).

For comparing the activities of the histidine-dependent DNAzyme and the hemin/G-quadruplex horseradish peroxidase-mimicking DNAzyme in the confined cavities with their activities in the homogenous solution, the activities of

the DNAzymes in the two different conditions were measured at the same concentrations of the active DNAzymes (24.6 nM for the histidine-dependent DNAzyme and 17.9 nM for the hemin/G-quadruplex horseradish peroxidase-mimicking DNAzyme). The time-dependent fluorescence changes of the products generated by the histidine-dependent DNAzyme and the hemin/G-quadruplex horseradish peroxidase-mimicking DNAzyme were measured at $\lambda_{em} = 518$ nm and $\lambda_{em} = 585$ nm, respectively.

**Switchable unlocking and locking of nanoholes in origami O.** The origami tile O (2 nM; in TAE buffer with 6 mM $Mg^{2+}$ and 5 mM $Na^+$) was irradiated under UV light ($\lambda = 365$ nm, 25 °C) for 5 min and then was added with the helper hairpins $H_3/H_4$ (10 nM for each one, 25 °C for 10 h) to generate the nanohole. For the reversible locking process, the anti-helper strands ($H_{3a}'/H_{3b}'/H_{4a}'/H_{4b}'$) (50 nM for each one) were added into the sample to remove the strands $H_3/H_4$ (25 °C for 2 h) and then the sample was irradiated with visible light ($\lambda > 420$ nm, 25 °C) for 10 min. For the second cycle of unlocking and locking processes, fivefold helper strands, and the anti-helper strands compared with the amount of the corresponding strands used in the forward steps were added into the sample, and the sample was irradiated under UV light ($\lambda = 365$ nm, 25 °C) and visible light ($\lambda > 420$ nm, 25 °C) for 5 min and 10 min for the two processes, respectively.

For the control experiments to unlock the tile O (2 nM), only the UV irradiation ($\lambda = 365$ nm) or the helper hairpins $H_3/H_4$ (10 nM for each one) was used.

Agarose electrophoreses of the locked and unlocked tiles O were performed in the TAE buffer with 6 mM $Mg^{2+}$ and 5 mM $Na^+$ (1%, 100 V, 1 h, 25 °C).

**Switchable catalysis in the nanocavity of origami tile $O_f$.** The tile $O_f$ (30 nM, 200 μL, in TAE buffer with 6 mM $Mg^{2+}$, 5 mM $Na^+$, and 100 mM $K^+$) was prepared for the measurements of the reversible catalysis. In the locked tile $O_f$, the strands $G_{1\times}$ were activated by adding $B_{GX-1}'/B_{GX-2}'$ (300 nM for each, $B_{GX-1}'/B_{GX-2}'$: $B_{GX-1}/B_{GX-2}$ = 5: 1) to remove the block strands $B_{GX-1}/B_{GX-2}$ (25 °C for 1 h). For unlocking of the tile $O_f$, the sample was irradiated under UV light ($\lambda = 365$ nm, 25 °C) for 5 min and then was added with the helper hairpins $H_3/H_4$ (300 nM for each one, 25 °C for 10 h) to generate the nanohole and form the G-quadruplexes in the cavity. To close the unlocked tile $O_f$, crown ether (CE) (60 mM, 1800 μL; CE: $K^+$ = 5: 1) and the block strands $B_{GX-1}/B_{GX-2}$ (3 μM, 150 μL, $B_{GX-1}/B_{GX-2}$: $G_{1\times}$ = 10: 1) were added into the sample to anneal from 30 °C to 10 °C for 2 h. Then the anti-helper strands $H_{3a}'/H_{3b}'/H_{4a}'/H_{4b}'$ (6 μM, 150 μL; $H_{3a}'/H_{3b}'/H_{4a}'/H_{4b}'$: $H_3/H_4$ = 5: 1) were added into the sample to remove the strands $H_3/H_4$ (25 °C for 2 h) and then the sample was irradiated with visible light ($\lambda > 420$ nm, 25 °C) for 10 min. The sample was centrifuged to remove the excess helpers/anti-helpers/CE (100 k NMWL, 3000 × g, 10 min, three times). In order to measure the catalytic activities of origami $O_f$ in locked and unlocked states, 37.5 μL of the sample in respective state (set at 20 nM) was treated with the hemin (300 nM, 5 μL), Amplex Red (2 mM, 2.5 μL), and $H_2O_2$ (50 mM, 5 μL), and the mixture was incubated at 28 °C for 10 min. The fluorescence spectrum of the product (Resorufin) was measured using the fluorescence spectrophotometer ($\lambda_{ex} = 571$ nm). It should be noted that the reversibility of the photochemical opening and closure of the cavities is presented as a proof-of-concept characterizing the system for two cycles. In principle, we find that the reversible switching of the system can be extended to additional cycles. Nonetheless, the addition of the fuel and anti-fuel strands for each cycle diluted the sample and the repeated opening/closure of the cavities introduced accompanying damaged origami structures that perturb the reversible functions of the system. We find that after four cycles the switching degree decreased to 50%.

**AFM imaging.** For the AFM measurements, 2 μL of the respective origami tile samples were deposited on the surface of the freshly peeled mica to adsorb for 5 min. The samples were imaged under tapping mode in an aqueous buffer using SNL-10 probes (Bruke, Multimode Nanoscope VIII). Imaging was performing with a spring constant of 0.35 N m$^{-1}$ or with a spring constant of 0.24 N m$^{-1}$ and at a tapping frequency of ~9 Hz in buffer. Both of the conditions yielded very similar images.

The statistical analyses of the different systems included three different methods as follows: (i) scanning and analyzing of arbitrary four different domains of the same origami droplet deposited on the mica; (ii) scanning and analyzing of arbitrary four domains of four different droplets of the same origami mixtures deposited on different mica supports; (iii) the preparation of at least two different samples of the respective origami structures and analyzing at least two arbitrary domains of the same origami sample. The different methods led to very similar statistical analysis results ± 3%.

In addition, the statistical analyses of the AFM images revealed some incomplete origami structures. These imperfect structures included ether damaged origami tiles or structures where the open cavities were questionable. These structures were included in Supplementary Tables 1–16 as imperfect structures.

## Data availability

All Data supporting the findings of this manuscript are available from the corresponding author upon reasonable request. The source data underlying Figs. 1c, d, h, i, and 3c, d, and Supplementary Figs. 6c, 20b, 21b, and 27 are provided as a Source Data file.

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

## Acknowledgements

Parts of this project are supported by the Israel Science Foundation and by the Minerva Center for Biohybrid Complex Systems.

## Author contributions

J.W. planned the origami tiles and performed the measurements. L.Y. participated in the design of the DNAzyme units and formulated the means to transduce the DNAzyme functions. Z.L., J.Z., and H.T. synthesized the azobenzene strands. I.W. supervised the project and participated in the planning of the origami nanostructures. All authors participated in the formulation of the paper.

## Competing interests

The authors declare no competing interests.
