## [Peer Review File · Nature Communications]

Reviewers' Comments:

Reviewer #1:

Remarks to the Author:

This manuscript seems overall quite well written, the experiments simple and elegant in design, and the corresponding interpretations relatively concise, straightforward and easy to understand. I honestly found it somewhat difficult to come up with any criticisms since to the best of my knowledge and understanding, everything looks quite well thought out. Consequently I managed to have only two minor comments and a few grammar and typo corrections.

1. While the AFM data and the corresponding statistics for the respective experiments were shown as a 4 panel figure, each panel being an image from a different scanned area, it would be interesting to know exactly which parameters/conditions these images differed in, i.e. for example are these 4 different scanned areas on the same mica surface i.e. the same replicate? Or are these from different replicates of the same experiment?

2. There is essentially no doubt regarding the results of all the experiments, in that there seem to be very clear differences between all the experiments and their corresponding control treatments. However it is just good scientific form to also show some preliminary statistic tests to accompany the claims.

Grammar and typo

1. L57 - "...chiroptical functions were demonstrated..." should be "chiroptical functions have been demonstrated"

2. L60 - "...functional origami structures included, however, the bottom-up..." should be "functional origami structures involved, however, the bottom-up"

3. L64 - "To date, such cavities were fabricated..." should be "To date, such cavities have been fabricated"

4. L65 - "...and the cavities were used for site-specific docking of antibodies, the reconstitution of membrane proteins..." should be "and these cavities were used for the site-specific docking of antibodies, reconstitution of membrane proteins..."

5. L67 - "In contrast to this previous art, the present study introduces a concept for the "active" fabrication of nanoholes in origami tiles." should be "In contrast, the present study introduces the concept of "active" fabrication of nanoholes in origami tiles."

6. L73 - "...we highlight the design of a reversible..." should be "we highlight a design for the reversible"

7. L294 - Typo - "nanohole-patters" should be "nanohole-patterns"

8. L 480 - Typo - "spectru" should be "spectrum"

Reviewer #2:

Remarks to the Author:

The manuscript by Wang et al. presents an exquisite DNA origami tile featuring a switchable catalytically active cavity.

A similar concept was presented by the authors in an earlier publication (Wang et al., Nano

Letters, 2018, <https://doi.org/10.1021/acs.nanolett.8b00793>). In this earlier publication, the active state of the DNAzyme (catalyzing a similar reaction) was triggered by the dimerization of two DNA origami subunits instead to the opening of a cavity within the origami.

The authors have done a considerable amount of work to demonstrate the opening/closing of the cavity upon application of different stimuli, but several points should be addressed before publication in Nature Communications or elsewhere:

Comments:

1) In the introduction, the authors state that their study represents the first example of an active generation of nanocavities. This is not true. To give a more balanced account of the literature, examples of actively switchable stimuli-responsive nanopores should be mentioned, e.g. voltage-dependent cavity opening by Seifert et al. (ACS Nano, 2015, <https://doi.org/10.1021/nn5039433>) or ligand-gated cavity opening by Burns et al (Nature Nanotechnology, 2016).

2) For a non-specialist reader, it would be beneficial if the authors could provide literature references when they first mention the used DNAzymes. More descriptive details on the catalytic reaction and the functioning of the DNAzyme could be helpful.

3) The authors repeatedly state percentages of locked versus open tiles. How were these values determined? Looking at the AFM images, some of the structures could be classified as either of the two categories. Did they set some threshold in the image analysis? Were non-intact structures disregarded in the analysis? This should be described in detail in the AFM section of the Methods part.

4) I could imagine that the surface absorption that is required for AFM imaging may introduce a systematic bias in the percentage of open tiles. Have the authors considered e.g. FRET to validate their results in solution?

5) Similarly, in Fig. 2, according to what metric were the tiles in the AFM image divided into "N" and "M"? To me, both structures look rather similar – the four-hairpin label on M can be mistaken for N in its open configuration. Did the authors perform a statistical analysis to exclude an unwanted bias?

6) Given that AFM imaging is used as the key method throughout the manuscript, the authors provide very limited details about the imaging parameters in the method section. E.g. it remains unclear which cantilevers (spring constant?) were used. This section has to be expanded.

7) The authors compare the activity of the DNAzyme in closed and open DNA origami cavities. How do the values compare to the activity of the bare enzyme? Does cavity opening fully restore its activity?

8) The authors demonstrate an "on-off-on-off" cycle to showcase the reversibility of their system (Fig. 4). Have they tested or can they estimate how many repeats are possible? This should be discussed.

9) Supplementary Figure 25: Please show the full gel with a reference ladder instead of a cropped version. Cropping makes it difficult for the reader to assess the result.

10) In the interest of reproducibility of research and open science, a complete list of DNA sequences has to be provided. The authors should also consider to share the design file for their DNA origami structure.

Response to referees

The following changes/explanation were introduced into the paper.

Reviewer #1:

I appreciate very much the very positive evaluation of the reviewer and his/her comment that “... .. *the experiment simple and elegant in design, and the corresponding interpretations relatively concise, straight forwards and easy to understand. I honestly found it somewhat difficult to come up with any criticism*”.

Answers to the specific comments:

1. “*While the AFM data and the corresponding statistics for the respective experiments were shown as a 4 panel figure, each panel being an image from a different scanned area, it would be interesting to know exactly which parameters/conditions these images differed in...*”

Reply: The issue of statistical analysis of the images was addressed in detail in the experimental section, heading *AFM imaging*, in the revised manuscript, page 20.

2. “*... it is just good scientific form to also show some preliminary statistic tests to accompany the claims.*”

Reply: We added the statistical analysis of the control systems and these are outlined in new tables: Supplementary Table 2, Supplementary Table 3, Supplementary Table 5, Supplementary Table 6, and Supplementary Table 15 and Table 16. These statistical analyses were mentioned in the revised manuscript, pages 6, 7 and 12, respectively.

3. “*Grammar and typo*”

Reply: All errors were corrected in the revised manuscript, pages 3, 4, 14 and 17, respectively.

Reviewer #2:

“*A similar concept was presented by the authors in an earlier publication (Wang et al., Nano Letters, 2018, <https://doi.org/10.1021/acs.nanolett.8b00793>). ...*”

Unfortunately, we disagree with this statement of the reviewer. The quoted reference significantly differs from the present work. In the quoted reference, we described the interconversion between two origami tiles using the strand displacement process. The present study introduces the DNAzymes and light as unlocking mechanisms of a domain on the origami tile and the development of an active “window-opening” mechanism using “handles”, “helper units” and anchoring footholds.

Answers to the specific comments:

1. “*In the introduction, the authors state that their study represents the first example of an active generation of nanocavities. This is not true. ...*”

Reply: The statement, although being correct, was not introduced into our original version of the paper. The references mentioned by the reviewer have little relevance to the present

paper. They describe the incorporation of nucleic acids into membranes and the generation of nanochannels for transport. These references do not deal with DNA origami structures. To follow the reviewer comment, we introduced the two references to the introduction and explained the differences between our work and the stated references in the revised manuscript, page 3.

2. *“For a non-specialist reader, it would be beneficial if the authors could provide literature references when they first mention the used DNAzymes. ...”*

Reply: We introduced a discussion on catalytic nucleic acids (DNAzymes) as requested in the revised manuscript, page 4. We described shortly their applications in DNA nanotechnology.

3. *“The authors repeatedly state percentages of locked versus open tiles. How where these values determined? Looking at the AFM images, some of the structures could be classified as either of the two categories. ...”*

Reply: In the experimental section, heading *AFM imaging*, we added a detailed explanation on the statistical analysis of the different structures in the revised manuscript, page 20. The yields of “non-intact” origami tiles and their effect of the statistical analysis of open/closed origami were introduced in the supplementary Table 1 to Table 16 (in supplementary information, pages 4, 7, 9, 14, 17, 19, 21, 23, 25, 36, 38, 40, 42, 44, 47 and 49, respectively).

4. *“... Have the authors considered e.g. FRET to validate their results in solution?”*

Reply: This was an excellent comment of the reviewer. The FRET experiment to validate the AFM imaging was performed. The experiment is described and explained in detail in Supplementary Figure 5. The FRET experiment indicates a 70% “window” opening yield, very similar to the yield derived from the AFM images. The results are, also, discussed in the revised manuscript, page 6.

5. *“Similarly, in Fig. 2, according to what metric where the tiles in the AFM image divided into “N” and “M”? ...”*

Reply: We explained that the tiles “N” and “M” can be distinguished by following the heights of the open window and/or of the hairpin markers in the revised manuscript, pages 8-9.

6. *“... which cantilevers (spring constant?) were used. This section has to be expanded.”*

Reply: Details on the spring constant and tapping frequency of the cantilevers were added to the *AFM imaging* heading, experimental section in the revised manuscript, page 19.

7. *“The authors compare the activity of the DNAzyme in closed and open DNA origami cavities. How do the values compare to the activity of the bare enzyme? Does cavity opening fully restore its activity?”*

Reply: This comment of the reviewer led to very interesting results. We find that the histidine-dependent DNAzyme and the hemin/G-quadruplex DNAzyme reveal ca. two-fold higher activities as compared to the DNAzymes in the homogeneous solution!! These interesting results are now presented in Figure 3(c) and (d), and discussed in the revised

manuscript, pages 10 and 11. The enhanced activities originate from the spatial concentrations of the catalytic units in the confined nanocavities.

8. “... *Have they tested or can they estimate how many repeats are possible? This should be discussed.*”

Reply: The photochemically-induced cyclic reversibility of the azobenzene-locked window system has been addressed in the revised manuscript, page 19.

9. “*Supplementary Figure 25: Please show the full gel with a reference ladder ...*”

Reply: The full electrophoretic gel of the origami structures, including the reference ladder, is presented in Supplementary Figure 26, page 50 in supplementary information.

10. “*In the interest of reproducibility of research and open science, a complete list of DNA sequences has to be provided. ...*”

Reply: The complete set of sequences was added to the supplementary material (Supplementary Tables 17-24). Paragraphs describing the design of the origami tiles with the locking/unlocking mechanisms are detailed in Supplementary Figure 1 (supplementary information, page 2), Figure 6 (supplementary information, page 12) and Figure 18 (supplementary information, page 34).

Reviewers' Comments:

Reviewer #1:

Remarks to the Author:

The authors seem to have addressed all concerns except perhaps those raised in #2.

It was mentioned that while the raw numbers clearly highlight the difference between all the experiments and their corresponding controls, it is good practice to perform at least an elementary statistical test. What was meant by this was something like even a simple Student's T-test, to quantitatively conclude that the experiment was statistically significantly different from its control (for example in this case, the locking/unlocking percentages in the presence/absence of helper strands, or presence/absence of ions). It is recommended that the authors perform such test(s).

Reviewer #2:

Remarks to the Author:

The authors have done a considerable amount of work (text modifications and experiments!) to address several critical points. The experiments led to interesting new insights which not only support the previous results but also lift the impact and novelty. The manuscript will, no doubt, be of high relevance for the broad readership of Nature Communications and I fully support its publication without further modifications.

Response to referees

Answer to the specific comment from Reviewer #1:

1. *“It was mentioned that while the raw numbers clearly highlight the difference between all the experiments and their corresponding controls, it is good practice to perform at least an elementary statistical test. What was meant by this was something like even a simple Student's T-test, to quantitatively conclude that the experiment was statistically significantly different from its control (for example in this case, the locking/unlocking percentages in the presence/absence of helper strands, or presence/absence of ions). It is recommended that the authors perform such test(s).”*

Reply: We add Supplementary Figure 5 that presents the statistically significant differences of the results according to Student's T-test, as requested by the Reviewer. In addition, the use of this statistical method to analyze the results presented in Supplementary Figures 2-4 and Supplementary Tables 1-3 was mentioned in the text.